# Lessons from the First Wave of COVID-19. What Security Measures Do Women and Men Require from the Hotel Industry to Protect against the Pandemic?

**DOI:** 10.3390/ijerph18052232

**Published:** 2021-02-24

**Authors:** Ramón Rueda López, Teresa López-Felipe, Virginia Navajas-Romero, Antonio Menor-Campos

**Affiliations:** 1Department of Statistic, Econometrics, Operational Research, Business Organization and Applied Economics, University of Cordoba, 14071 Córdoba, Spain; ramon.rueda@uco.es (R.R.L.); vnavajas@uco.es (V.N.-R.); 2Department of Agrarian Economy, Finance and Accounting, University of Cordoba, 14071 Córdoba, Spain; es1mecaa@uco.es

**Keywords:** COVID-19, tourism, hotel sector, security measures, sex differences

## Abstract

The tourism sector in general and the hotel sector in particular face the challenge of managing appropriate security measures to deal with the COVID-19 pandemic. In this sense, it is useful to know which measures are most demanded by the clientele. This research, through non-parametric statistics tests, concluded that women are more demanding than men in relation to the security measures to be taken in hotels. More specifically, this research concludes that women are more demanding than men in relation to a set of measures including ensuring good hygiene conditions, the use of disinfectants, the existence of health and information checks, adapting the establishment to WHO recommendations, obtaining quality certification, measuring temperature, the need to provide information on protocols and measures, and the elimination of physical contact between people. This, as a practical application, makes it possible to know more accurately about the safety requirements of sex-segmented customers in the face of future health crises, allowing tourist managers to offer safer destinations and the hotel sector better health conditions for their clients.

## 1. Introduction

On March 11, 2020, the World Health Organization (WHO) declared a global pandemic caused by the SARS-CoV-2 coronavirus strain [1]. This pandemic spread globally within a few months, affecting economic and social aspects around the world. Travel bans and social estrangement are recurrent public health guidelines in pandemic containment and have had a major impact in industries with high levels of human interaction or “high contact” [2], such as hotels or other tourist activities, hard-hit in this period [3]. However, the pandemic of the COVID-19 is not the first health crisis affecting travel in particular and tourism in general. In recent decades, other viral epidemics such as severe acute respiratory syndrome (SARS), also known as SARS-CoV-1, MERS, swine flu, Ebola, Zika, or yellow fever have also threatened public health around the world [4]. Unlike health problems such as the 2003 SARS outbreak or the Ebola crisis of natural disasters [5,6,7,8,9] or of social revolts such as the Arab Spring [10], the COVID-19 pandemic is a reality. However, according to Lori Pennington-Gray, director of the Tourism Crisis Management Initiative, it is the first time that a health crisis (or of any other type) has become a global crisis and is affecting all countries of the world and all facets of tourism activity [11].

Paradoxically, tourism activities, and especially travel, have become a vehicle for the diseases likely to become pandemics; therefore, tourism is the sector most affected by actions designed by the public health authorities for the mitigation of the pandemic [9]. The connection between tourism services and the risk of health disasters has forced governments to restrict, or even prohibit, travel as a measure for managing the risks posed by the transmission of the virus [1,9].

Air transport is currently the main means of pandemic spread [12]; hence, the measures taken to minimize SARS transmission between March and May 2003 focused on reducing passenger numbers at the major airports concerned. This reduction was between 57% and 77% [12].

This has happened in other health crises. In 2009, there was a reduction in passenger volume of between 4.12 and 7.88% due to swine flu. However, as a differentiating element, in the case of avian influenza, in 2006, passenger volume in affected areas increased by 9.04% to 16%. In these three previous pandemics, measures such as taking the temperature of travellers were adopted at airports, containing travellers with symptoms. In fact, the responses given by countries in the current pandemic have followed protocols similar to the authors [13,14]. However, there are aspects that differentiate the current pandemic from the previous ones in terms of its effects and the scope and effectiveness of the measures taken to contain them, such as the life cycle of pathogens caused by quarantine situations. According to the experts, the lack of preparation of the tourism industry in the current situation could be explained by the differences in the COVID-19 pandemic in relation to those mentioned above since these did not have a significant impact on the decline in international travel [9]. On the other hand, the incubation period of the other pandemics was shorter and more noticeable compared to COVID-19’s, which has made it sometimes undetectable. That characteristic has facilitated its spread via travellers [15]. 

Specifically, within the tourist sector, the hotel is one of the most vulnerable to the effects of the COVID-19 pandemic [16,17].

In Spain, according to data published by the National Statistics Institute (NSI)—the independent administrative autonomous institution assigned to the Ministry of Economic Affairs and Digital Transformation—overnight stays in hotel establishments decreased by 78.4%, the average stay for each visitor was shorter by 31.28%, and the occupancy rate at bed-places was 60.45% lower than the same month of 2019 [18].

In this regard, as Shin and Kang [19] point out, there is a high perception of the risk of destinations and hotel properties. For this reason, it is necessary to address, within the hotel industry, the study of the necessary actions that can defuse the effects of the pandemic on the hotel sector. As Jiang and Wen [20] show, it is appropriate to define strategies and actions that promote the confidence of guests, supporting the recovery of the sector. However, a scenario is possible where the hotel industry can be more resilient and sustainable from the security needs of guests, adapting them through measures that can transform threats into opportunities [20].

Finally, the adjustment of the expectations placed by the World Tourism Organization (UNWTO), from predicting on 6 March 2020 a two to three percent reduction in international travel is significant, compared to the 2019 figures, to a 20 to 30% reduction announced by 26 March 2020 [9], which in subsequent months was increasing.

The objective of this research is to present the results of a study carried out in Spain to identify the perceptions and opinions that women and men have with regard to tourist destinations that are marked by this pandemic.

Research on gender differences in the tourism sector has been conducted from various points of view, such as the image of the destination and travel options [21], the commitment and loyalty of the visitors [22], the risks [23], and, more specifically, in tourist areas such as golfing locations [24], archaeological sites [25], or World Heritage sites [26].

Moreover, recent research has addressed the impact of COVID-19 on tourism. These studies include those of Roman et al. [27], who confirmed the great impact of the COVID-19 pandemic on the organization of tourism travel during 2020; Hong et al. [28], who have studied the impact of the pandemic on the bed and breakfast (B&B) tourism industry in China; or that of Han et al. [29], who revealed that corporate social responsibility in the tourism sector improves attitudes and behavioural intentions of passengers or guests.

In our case, this investigation, in particular, aims to check whether there are differences between women and men in terms of the safety measures required by each group to deal with the hotel sector in order to deal with the COVID-19. This will be analysed through a segmentation by gender which will enable a better understanding of the needs and expectations of both population groups in reaction to the measures required to protect tourists from COVID-19 in the hotel sector.

Achieving this objective makes it possible to better understand the expectations that women and men have separately in terms of security. Something relevant if the rational decision-making process for the purchase of tourist goods and services is considered to be different between women and men [30]. Women, in this regard, are often more sophisticated consumers, paying greater attention to details [31,32].

The findings and conclusions that can be obtained from this research may be of particular interest for the hotel sector, allowing managers of these establishments to design and manage security measures that make their guests feel safer in future waves of the pandemic, thus obtaining a return of women’s opinions regarding the risks perceived during travel [23].

To achieve this objective, this research is organised as follows: First, we present backgrounds on the impact and consequences of COVID-19 on tourism, on the possibility and challenges identified to address the pandemic in the tourism sector, the way in which differences between women and men in the tourism sector have been addressed, and the safety measures to be taken against COVID-19 in the tourism sector; these backgrounds will serve to define the hypothesis of the research. Next, we present the methodology of the investigation. This is followed by a section in which the results of the investigation are presented and analysed and, finally, the conclusions paragraph summating the findings of the investigation will be presented.

## 2. Background

### 2.1. On the Impact and Consequences of COVID-19 on Tourism

The impact of COVID-19 and its consequences on tourism depend on factors marked by uncertainty, which can be grouped into three categories: first, the duration of the crisis (pandemic control, travel restrictions, reactivation of transport, control of successive waves of COVID-19, etc.); second, the support policies that governments implement (who the beneficiaries will be, how effective these policies will be, etc.); and third, the kind of tourist behaviours that arise (if consumers decrease or intensify their desire to travel; the role that confidence among tourists will play in nations or areas as a tourist destination, how health and safety are perceived, etc.). These three categories of external factors are complemented by situational factors in each of the destinations, such as dependence on tourism, proper governance of the destination at different levels, or the willingness to adapt to different tourist behaviour [33].

There are also voices that say nothing will ever be the same again in tourist activity and the high likelihood that the socio-economic changes produced by this pandemic will have a very significant impact on tourism in aspects such as mobility, patterns of socialization and consumption, or the relationship between leisure and work [34]. The UNWTO estimated a decrease of between 20 and 30% in international tourist arrivals and corresponding economic income in 2020 compared to 2019. However, the UNWTO recognizes that such estimates should be treated with considerable caution, given the magnitude, volatility, and completely different profile of this health crisis compared to previous pandemics [35].

### 2.2. On Possibilities and Challenges to Face the Pandemic

The pandemic of the COVID-19 virus is causing a huge problem for tourism activity in all countries and it is unknown how the tourism sector will develop afterwards, in a period of economic and social crisis. Therefore, the key question, which is the objective of this research, is what tourists will demand in terms of safety measures when they can travel again, and for this question, there are several answers that are related to the transformation that the COVID-19 pandemic will provoke in tourist activity.

From the scientific research perspective, the transformative potential of the pandemic towards sustainability is emphasized, reconsidering a global tourism system more aligned with the United Nations Sustainable Development Goals (SDG) [8,9,36]. Thus, the importance of a tourist model aligned with the SDG should be focused on sustainability and people as a question of coherence and justice, with a technical and political basis in line with the achievement of the SDGs and to mark the roadmap to the new future of tourism [37]. 

The post-COVID-19 tourism model must have a developed learning capacity and be able to anticipate future crises with new priorities and needs of the population and tourists [38].

Sustainability refers to how specific models of ecological partner systems respond to disturbances [39]. It tries to respond to how tourism can adapt to the social, political and economic change that is causing this pandemic [40]. 

The UNWTO’s approach [35] is broader and concerns the SDGs and resilience along with institutional strengthening to mitigate the impact of the health crisis and accelerate economic and social recovery. This recovery is based on the following resilience principles set out by Biggs et al. [41] and applied by Berbes-Blazquez and Scott [42]: 

(a) Diversity and redundancy: different types of attractions, different groups of objectives to combat vulnerability and overcrowding, organization of events, etc.

(b) Connectivity: the links of a tourism system, which requires networks at various levels from aviation for international tourism to land transport for local tourism.

(c) Management of slow variables and feedback: slow variables report on the dynamics of a system, such as the diverse and changing preferences of tourists, while fast ones do so about the flow of tourists.

(d) Experimentation and learning: development of new forms of tourism and innovative offers, tourism destination management teams to promote research and the learning of tourist patterns. 

(e) Central participation and governance: meeting of the various stakeholders, citizens and experts, to improve decision-making, and the independence of units acting with vertical and horizontal links.

The aftermath of COVID-19 in the tourist activity anticipates the emergence of new tourist consumption habits and patterns based on greater social and environmental awareness [6,43], which will show consumer concern about sustainability and social problems, both nationally and internationally [43,44,45,46,47]. Consequently, post-COVID 19 tourists will choose to travel to destinations that are closer to their place of residence and safer. 

In the context of insecurity and uncertainty, nearby destinations could be considered “less risky” by many potential tourists who, having been remarkably affected by the social and economic crisis arising from the health crisis, have seen their purchasing power reduced. Added to this are restrictions on international (long-distance) travel, at least for a while, to help reduce air pollution, which would undoubtedly be in line with the promotion of more sustainable tourism and the development of the concept of decline. In this regard, greater adaptation to future pandemics of those companies that adapt to the expected change in the consumer, which will include a greater demand for sustainable tourism, is expected [43]. This implies that tourism companies must be strongly rooted in the destination, that have been loyal to the principles of sustainable tourism, that offer ecotourism products or based on the local natural and cultural heritage, that provide high-quality experiences for tourists and that achieve an added value for the destination and, logically, for the local community itself, and all of this with maximum health safety.

As a result, a paradigm shift for post-COVID-19 tourism will replace the market share approach with one based on value-sharing and where the marketing of tourism companies has to be oriented to listen to what people want and are passionate about in order to share and satisfy those desires [48]. 

The challenges that arise in the post-COVID-19 world challenge tourism stakeholders to develop transition plans with scenarios incorporating sustainability, resilience, and internationalization [33]. Therefore, the work that arises on the role of robotics and the development of technology in the adaptations necessary to deal with the safety and hygiene measures imposed by the pandemic in airports, shopping centres, tourist accommodation, or restaurants [49] is necessary. Also necessary is scientific research on the adaptation of tourist cities with models and techniques based on robotics and computing, and that represent the context of a particular application of resilience [50], estimating the spread of COVID-19 in meaningful urban contexts [51]. It will also be necessary to take into account the implications of the pandemic on information systems and in the proactive collection of data for knowledge of future tourist demand based on the use of big data and not on historical data [52].

### 2.3. On the Differences between Women and Men in Tourism

Sex is an important variable to consider in the behaviour and ways in which consumers make their decisions [53]. In this regard, it is appropriate to visualize the views that women, as a group, may have, since they are increasingly important as consumers in general [54] and as consumers of tourist products in particular [31]. In relation to this, men make decisions more quickly and intuitively, while women take into consideration the views of their families and friends [55] and, as Karatsoli and Nathanail pointed out [56], are influenced by their social networks.

However, taking as a reference the sex variable to identify differences in relation to the motivations, the experiences or satisfaction of tourists who, for example, visit a heritage site of humanity, creates dispute.

Research such as that of Adie et al. [57], Chen and Huang [58], and Correia et al. [59] identified no sex differences, while the research of Ramires et al. [60], Wang et al. [61], and Huete-Alcocer et al. [62] did, noting that women are more willing to visit heritage destinations. On the contrary, Wang and Hao [21] argue that it is men who have the greatest predisposition to visit historical sites.

However, sex differences regarding measures required by the tourist sector in general and the hotel sector in particular to address COVID-19 have not been thoroughly investigated.

For a city that, for example, bases its tourism on historical and cultural heritage, knowing these differences between sexes is relevant for two reasons.

First, because the rational decision-making process for the purchase of tourist goods and services is different between women and men [30]. Women are usually more sophisticated consumers and pay greater attention to details [31,32]. For this reason, it seems appropriate to determine what differences between sex may be in order to enable local tourism plans and the hotel sector to provide preventive measures against COVID-19 that meet the expectations of women and men separately [26].

Second, according to Wang and Hao [21], women have less of a predisposition than men in the choice of heritage tourist destinations. Therefore, analysing separately what the most-demanded security measures are for women and men in relation to COVID-19 may be important in providing a safer and more attractive tourist destination.

Thus, under the aim of the research and taking into account the revision of literature, the hypothesis that can be drawn is hypothesis 1) The level of requirements for security measures in hotel establishments is different between women and men.

### 2.4. On Security Measures in Tourism against COVID-19

The pandemic has paralyzed tourism globally and its relaunch at the so-called “new normal” stage requires a balance between maintaining a satisfactory experience for tourists and complying with the strict measures taken by the authorities on safety and hygiene to ensure a reactivation of the tourism industry once the containment phase has passed. Thus, in mid-May, the World Travel and Tourism Council (WTTC) presented the global protocols for the revival of the tourism sector with the claim of building consumer confidence, recovering jobs, and compensating for the financial losses caused by the fall in tourism worldwide. The protocols capture measures that have been designed by all industry representatives worldwide and are based on medical evidence, standards established by the World Health Organization (WHO) and the Centers for Disease Control and Prevention of the U.S. Department of Health & Human Services (CDC). They constitute a means of approval of criteria and provide health guidance to suppliers, travel operators, and tourists.

Regarding tourist accommodation, and specifically hotels, these protocols and recommendations have been the subject not only at a global scale but also by the European Commission, through the communication for the progressive restart of tourism services and health protocols in hospitality establishments (2020/C 169/01), published in the Official Journal of the EU of May 15, 2020 [63], and of national governments; in the case of Spain, it has been formalized in a document prepared by the Institute for Spanish Tourism Quality [64] in coordination with the Secretary of State for Tourism, the Autonomous Governments of the Spanish regions and the Spanish Federation of Municipalities and Provinces. According to all of them, the measures and recommendations designed for their development and application in hotel establishments have generally been realized in processes of deep cleaning and hand washing, between staff and customers, and use of protective equipment (masks, gloves and other protective measures); cleaning and disinfection of common spaces and contact points such as railings, tables, handles, sinks, etc.; disinfection of room cards, TV controls, light switches, and thermostats as well as the promotion of electronic payment; installation of alcohol-based hand-sanitizer dispensers on each floor, in entrances and outlets; cleaning and reduced capacity in elevators; as well as encourage the use of stairs; in-room breakfast delivery, if possible, and a guarantee in buffets that guests do not handle food. They also point to the importance of assigned seating plans in common areas to avoid physical contact and agglomerations, as well as clear information and signage; limitation of capacity, social distance and identification of risk zones in entrances and points of greater influx; completion of risk assessment questionnaires before accessing the establishment; reservation of isolation spaces in the hotel itself for users presenting symptoms of COVID-19 during their stay. 

In relation to the above measures, it is worth noting the study carried out in hotels in the Canary Islands [65] whose results on the changes that the pandemic has brought in tourist accommodation reveal that the protocols implemented will be sufficient for the reopening of tourism, that the greatest adaptation efforts are concentrated in the common areas, areas of restoration and cleaning of rooms. The most important measures related to common areas are social estrangement and the use of personal protective equipment (PPE) by workers, while major re-opening investments focus on protective equipment and signage to organize customer transit, followed by the placing of protective screens.

The proposed objective for this research is to determine the extent to which sex differences exist in terms of measures required against COVID-19 by tourists in the hotel sector; therefore, taking into account the revision of the literature carried out in the contrast scenarios, the hypotheses are as follows:

**Hypothesis** **1.**
*The level of demand for respecting social distance in hotels is different between women and men.*


**Hypothesis** **2.**
*The level of requirement regarding the good hygiene conditions that a hotel must present is different between women and men.*


**Hypothesis** **3.**
*The level of demand for the information that hotel employees must have in front of COVID-19 is different between women and men.*


**Hypothesis** **4.**
*The level of demand for avoiding physical contact within a hotel is different between women and men.*


**Hypothesis** **5.**
*The level of demand for using the mobile phone to register at the hotel is different between women and men.*


**Hypothesis** **6.**
*The level of demand for considering the importance of hotels being “immune” is different between women and men.*


**Hypothesis** **7.**
*The idea that it is better to stay in small hotels is different between women and men.*


**Hypothesis** **8.**
*The idea that the hotel should deliver a virus prevention kit is different between women and men.*


**Hypothesis** **9.**
*The idea that the use of QR codes should be extended in hotels is different between women and men.*


**Hypothesis** **10.**
*The idea that disinfectants such as ozone should be used in hotels is different between women and men.*


**Hypothesis** **11.**
*The idea that there should be health checks in hotels is different between women and men.*


**Hypothesis** **12.**
*The idea that official information should exist in hotels is different between women and men.*


**Hypothesis** **13.**
*The idea that hotels should adapt to World Health Organization (WHO) recommendations is different between women and men.*


**Hypothesis** **14.**
*The idea that hotels should have a quality certification for coronavirus prevention and control is different between women and men.*


**Hypothesis** **15.**
*The idea that hotels should measure the temperature of customers is different between women and men.*


**Hypothesis** **16.**
*The idea that hotels should inform customers about protocols and measures is different between women and men.*


**Hypothesis** **17.**
*The idea that hotels should eliminate physical contact between staff and customers is different between women and men.*


## 3. Methodology

### 3.1. Data Collection

The methodology used in this research is based on the realization of a field work to a representative sample of people in Spain to know their opinions and perceptions of what tourism will look like after the pandemic.

The data collection process was conducted through an online survey disseminated by general social networks and specialised in tourism. Fieldwork was carried out between April and June 2020. This research collected a total of 332 responses, of which 328 were valid. 

A non-probabilistic technical sampling technique was used, commonly employed in this type of research where interviewees are available for surveys in a given space and time [66]. The survey, as it was carried out online, addressed, in particular, those people who are active in social networks specialized in tourism and travel. No stratification of the sample was carried out by age, education, nationality, or by any other variable as there were no previous studies that supported this stratification.

To test the reliability of the scale, Cronbach’s Alpha test was performed, yielding a value of 0.830, a value above the minimum limits of 0.7 set by Nunnally and Bernstein [67].

### 3.2. Survey Questionnarie

The quantitative methodology used in this research has been based on a questionnaire based on previous studies [65,68,69,70]. The questionnaire was completely anonymous and was divided into three clearly differentiated blocks. The first block addressed questions related to the respondent’s way of making the trip. The second of the blocks addressed aspects that looked at the measures taken in hotels and how this pandemic can affect the tourist experience in a given destination. The third block addressed the sociodemographic profile of respondents where aspects such as sex, age, level of study or household income were analysed. 

The questions included within the second block were asked through five-point Likert scales, where 1 referred to “Very much disagree,” 3 “Neither disagree nor agree” and 5 “Very much agree”. The questions included in the first and third blocks were closed.

As noted above, this research is specifically intended to check whether there are differences between women and men in the safety measures that each group requires the hotel sector to deal with COVID-19. To achieve this objective, only the responses obtained in the second and third blocks of the survey have been taken into consideration. Appendix A presents the question that has been used in this investigation as a contribution to further investigations. 

### 3.3. Data Analysis

By calculating the means that both women and men have given to each answer, it is possible to accept or reject the research hypotheses. In this regard, the interest of research, as already justified, is to identify differences between women and men. Incorporating other study variables such as the level of studies, income level or country of origin, will remain for future research addressing a multivariant-type analysis.

Both for the preliminary analysis of data, processing and tabulation, as well as the corresponding statistical analysis, used the SPSS V.25 statistical software (IBM Corporation, Armonk, NY, USA) [71].

## 4. Results and Discussion

### 4.1. Sociodemographic Profile

According to the data presented in Table 1, it can be concluded that the profile of the model person who has participated in this study is that of a young woman residing in Andalusia (Spain) who is a full-time employee or student, and has an income that can range from 1000 to 2500 euros per month.

### 4.2. Descriptive Analysis

From each of the defined security measures, a safety requirement indicator (Table 2) has been developed, which can measure the overall level of requirement for the security measures demanded and with which hypothesis 1 can be contrasted.

From each of the defined security measures, a safety requirement indicator (Table 2) has been developed. This indicator has been obtained by the statistical transformation and recording of all security measures for which the persons were interviewed on a single scale of measurement such as the safety requirement indicator (SRI). The SRI has a dual interest in this investigation: on the one hand, it allows the overall level of demand for security measures demanded in both women and men or, in other words, how demanding hotel guests are in relation to security measures; moreover, it will be possible to accept or reject hypothesis 1.

While the SRI allows, in a holistic way, to understand the level of demand of women and men in relation to the level of safety required against COVID-19, and whether there are differences between the sexes, at the same time, proposing a set of 17 individual hypotheses allows us to know in depth the safety measures in which women and men are most demanding and whether there are differences between the sexes.

Table 3 lists, disaggregated by sex, the main descriptive statistics of the indicator of the security requirement. For this indicator, women are found to have an average value higher than men. Something that can be put in relation to the fact that women are more sophisticated consumers and pay greater attention to details than men [30,31]. However, this difference shall be checked by the corresponding statistical test in the following subsection.

On the other hand, Table 4 shows the main descriptive statistics for each of the security measures (SM) identified in this research.

A first assessment makes it possible to verify that all safety measures, except for SM7, obtain a valuation average of more than 3.5. In the case of the SM7, a measure that assesses the possibility of considering small hotel accommodation as safer; it obtains, in both men and women, the lowest of the average ratings—even less than 3.

Table 4 also lists the classification, according to the average assessment, of the safety measures that are considered most important according to sex. In each and every security measure, women have made higher assessments than men. If applicable, the first seven security measures they value score above 4.5. For their part, men only place two safety measures with scores above 4.5.

### 4.3. Analysis of Differences and Hypothesis Contrasts

In the absence of normality in the distribution of variables, contrasted with a 95% confidence level through the Kolmogorov–Smirnov test (Table 5), and assuming the heteroscedasticity of the variables, the non-parametric Mann–Whitney U test has been used [72] for the contrast of the hypotheses.

Table 6 shows the results of the Mann–Whitney U test for the SRI variable, and it is noted that there is a significant difference, with a 95% confidence interval between the averages of the indicator.

Therefore, regarding hypothesis 1, this hypothesis is accepted, stating that there are differences in the means; in other words, women are more stringent in terms of the measures to be taken in hotel establishments, which also reinforces the idea that women are more demanding and detailed tourist consumers than men [31,32].

Otherwise, Table 7 shows the results of the Mann–Whitney’s U test for the contrast of Hypothesis 1 to hypothesis 2.17 at a 95% confidence level, and which of these hypotheses can be accepted or rejected.

As mentioned above, this investigation was based on the idea that women are more demanding and detailed tourist consumers than men [31,32]. This idea can be reinforced by saying that women, in addition to being more demanding than men in terms of the safety measures to be taken in hotel establishments in relation to COVID-19, are also more demanding in terms of different measures, such as good hygiene conditions, the use of disinfectants such as ozone, the existence of health checks and official information, adapting the establishment to WHO recommendations, obtaining quality certification for coronavirus prevention and control, measuring the temperature of customers, as well as the need to provide information to customers on protocols and measures and, finally, elimination of physical contact between people.

To affirm, therefore, this is relevant, as mentioned in the introduction of this investigation, if it is assumed that the rational decision-making process for the purchase of tourist goods and services is different between women and men [30].

## 5. Conclusions

The COVID-19 pandemic is causing radical changes in consumer behaviour in 2020. In fact, one of the main sectors affected has been tourism activity, and especially hotel establishments. These establishments have had to adapt to respond to this pandemic. As a result, hotel establishments have strengthened their hygienic conditions.

This article presents the results of research conducted in Spain to learn how COVID-19 will affect the perceptions and experiences of post-COVID-19 tourists.

To analyse these results, it has been segmented by sex. As has been shown, women are more demanding than men in terms of the security measures they demand from the hotel sector. This can be found by the Safety Requirement Indicator (SRI) which reaches an average value of 4.744 for women and 4.5 for men. In addition, it is observed that women are also more existent than men in specific security measures such as SM1, SM2. SM3, SM4, SM6, SM10, SM11, SM12, SM13, SM14, SM15, SM16 and SM17.

These data have important practical applications. Those who, at first instance, enable the hotel establishment to be known as the security measures that are most concerned by their guests according to their sex. This is particularly relevant not only to provide safer accommodations, but also by finding that women are more demanding than men; it enables, to some extent, to influence women’s decisions as to which accommodation they choose. Secondly, these data may be useful for tourist managers to make destinations safer.

In turn, this research makes a contribution to scientific iteration by providing statistical elements that may be of relevance to conducting this study in other countries or cities where tourism is a major economic resource. It is ultimately to contribute, through statistical instruments, to enable the tourist sector in general and the hotel sector in particular to be safer against the pandemic anywhere on the planet.

As to the limitations of this research, we could consider the following: First, the field work for data collection was carried out between April and June 2020, the period in which the first wave of COVID-19 is dated. Currently, this wave may be of no interest; however, this research has tried to obtain adequate conclusions to face the second wave of COVID-19 that the world is currently going through.

Second, the comparison of media leads to limited conclusions; in this regard, a greater number of surveys would have enabled people to be segmented in such a way as to have been possible to study their particular characteristics, such as the level of studies, to determine how far other variables besides sex condition the requirement of safety measures. Third, the field work was also carried out in April and June 2020, i.e., during the first wave of the COVID-19; once the pandemic passes through its second wave, it is appropriate to note how far the data obtained have been able to vary. Moreover, more surveys would have made possible more relevant data. Fourth, this research only analysed demand and not supply. The analysis of the supply would have made it possible to note how far the safety measures taken by the hotel sector are in line with the security measures required by its guests, thus making it possible to adapt supply with demand.

These limitations open up the possibility of future investigations. In this sense, the research team intends to conduct a study in the coming months that could demonstrate the evolution of the demands of tourists after successive waves of the pandemic. At the same time, the safety measures taken effectively in hotel facilities would enable it to determine the extent to which the hotel supply satisfies the safety expectations of its clients.

## Figures and Tables

**Table 1 ijerph-18-02232-t001:** Socio-demographic profile.

Variable		%	Variable		%
Sex N = 266	Men	40.60%	Professional activity N = 269	Liberal professional	7.43%
	Women	59.40%		Entrepreneur	4.46%
Age N = 328	Less than 20	35.37%		Public servant	22.68%
	Between 20 and 29	24.70%		Full-time employee	24.16%
	Between 30 and 39	10.67%		Part-time employee	4.09%
	Between 40 and 49	8.84%		Self-employed	3.72%
	Between 50 and 59	16.16%		Student	25.28%
	More than 6	4.27%		Unemployed	5.95%
Region N = 262	Andalusia	66.79%		Retired	1.12%
	Community of Madrid	7.63%		Household work	1.12%
	Basque Country	4.58%	Education level N = 268	Primary	2.24%
	Outside Spain	4.58%		Secondary Education	19.40%
	Extremadura	3.82%		University degree	41.42%
	Castilla La Mancha	2.67%		Postgraduate/Master’s/PhD	36.94%
	Catalonia	2.29%	Income (€/month) N = 262	Less than 700 euros	4.96%
	Asturias	1.91%		From 700 to 1000 euros	11.83%
	Canary Islands	1.53%		From 1001 to 1500 euros	24.81%
	Castilla y Leon	1.15%		From 1501 to 2500 euros	25.19%
	Community of Valencia	1.15%		From 2500 to 3500 euros	16.41%
	Others	1.91%		More than 3500 euros	16.79%

Source: own elaboration.

**Table 2 ijerph-18-02232-t002:** Security measures and indicator.

Measures		Indicator
SM1	In the hotel you must respect the social distance.	Safety Requirement Indicator (SRI)
SM2	The hotel must have good hygiene conditions.	
SM3	Employees are well-trained against COVID-19.	
SM4	Physical contact with employees should be avoided.	
SM5	You must use your mobile phone to check into the hotel.	
SM6	It is important that hotels are “immune”.	
SM7	It is better to stay in small hotels.	
SM8	The hotel must deliver a virus prevention kit.	
SM9	In hotels, the use of QR codes should be encouraged.	
SM10	Disinfectants such as ozone should be used in hotels.	
SM11	There must be health checks.	
SM12	There must be official information.	
SM13	The hotel must be adapted to the recommendations of the World Health Organization (WHO)	
SM14	The hotel must have a quality certification for COVID-19 prevention and control.	
SM15	Customers’ temperatures must be measured.	
SM16	Customers should be informed about protocols and measures.	
SM17	Physical contact between people must be eliminated.	

Source: own elaboration.

**Table 3 ijerph-18-02232-t003:** Descriptive statistics.

Variable	Men					Women				
	N	Av	SD	As	Ct	N	Av.	SD	As	Ct.
Safety Requirement Indicator (SRI)	104	4.500	0.5912	−0.719	−0.436	156	4.744	0.4666	−1.506	1.186

Notes: Average (Av), Standard deviation (SD), Asymmetry (As), Curtosis (Ct). Source: own elaboration.

**Table 4 ijerph-18-02232-t004:** Descriptive statistics (Classification of measures by sex).

Men							Women						
Variable	N	Av	SD	As	Ct	Rk	Variable	N	Av	SD	As	Ct	Rk
SM2	109	4.862	0.499	−5.270	34.617	1	SM2	158	4.981	0.137	−7.117	49.269	1
SM13	109	4.587	0.735	−2.015	3.932	2	SM13	158	4.829	0.440	−2.624	6.458	2
SM1	109	4.459	0.776	−1.252	0.661	3	SM16	158	4.709	0.611	−1.949	2.489	3
SM12	109	4.404	0.992	−2.106	4.410	4	SM1	158	4.690	0.574	−1.705	1.896	4
SM3	109	4.395	0.933	−1.846	3.518	5	SM12	158	4.684	0.629	−2.432	7.748	5
SM16	108	4.278	0.874	−1.348	2.215	6	SM3	158	4.627	0.753	−1.892	2.436	6
SM4	109	4.202	0.911	−1.088	0.819	7	SM14	158	4.544	0.834	−2.178	5.182	7
SM11	108	4.194	0.981	−1.189	0.965	8	SM4	158	4.475	0.879	−1.943	3.867	8
SM6	107	4.084	1.158	−1.131	0.409	9	SM11	158	4.437	0.913	−1.794	3.029	9
SM14	109	4.073	1.069	−1.168	0.846	10	SM6	158	4.411	0.945	−1.735	2.595	10
SM9	109	3.917	1.123	−0.713	−0.428	11	SM10	158	4.095	1.199	−1.219	0.565	11
SM5	109	3.881	1.238	−0.874	−0.245	12	SM9	157	4.089	1.094	−1.101	0.541	12
SM10	109	3.670	1.210	−0.551	−0.561	13	SM15	158	4.089	1.186	−1.195	0.507	13
SM15	109	3.661	1.256	−0.643	−0.538	14	SM5	157	4.000	1.155	−0.886	−0.076	14
SM8	108	3.565	1.210	−0.558	−0.451	15	SM17	158	3.994	1.159	−0.882	−0.195	15
SM17	109	3.523	1.237	−0.398	−0.882	16	SM8	158	3.772	1.246	−0.738	−0.397	16
SM7	109	2.817	1.172	0.154	−0.484	17	SM7	157	2.975	1.245	0.089	−0.721	17

Notes: Average (Av). Standard deviation (SD). Asymmetry (As). Curtosis (Ct). Ranking (Rk). Source: own elaboration.

**Table 5 ijerph-18-02232-t005:** Kolmogorov–Smirnov test.

Variable	N	Test Statistic	Sig. Asymptotic (Bilateral)
SRI	285	0.421	0.000
SM1	295	0.421	0.000
SM2	297	0.526	0.000
SM3	296	0.405	0.000
SM4	296	0.331	0.000
SM5	296	0.268	0.000
SM6	295	0.347	0.000
SM7	294	0.207	0.000
SM8	295	0.199	0.000
SM9	294	0.266	0.000
SM10	294	0.265	0.000
SM11	292	0.341	0.000
SM12	295	0.405	0.000
SM13	295	0.463	0.000
SM14	294	0.350	0.000
SM15	294	0.254	0.000
SM16	294	0.406	0.000
SM17	294	0.236	0.000

Source: own elaboration.

**Table 6 ijerph-18-02232-t006:** Mann–Whitney U test for Hypothesis 1.

Variable	Test Statistics	
SRI	U	5816.0
	Sig. asymptotic (bilateral)	0000

Source: own elaboration.

**Table 7 ijerph-18-02232-t007:** Mann–Whitney U test; and Hypothesis accepted and rejected.

Security Measure	Mann–Whitney U test		Hypothesis	Accepted or Rejected
	U	Sig. Asymptotic (Bilateral)		
SM1	7353.0	0.012	Hypothesis 1	Accepted
SM2	7902.5	0.003	Hypothesis 2	Accepted
SM3	7304.0	0.009	Hypothesis 3	Accepted
SM4	6925.0	0.002	Hypothesis 4	Accepted
SM5	8136.5	0.468	Hypothesis 5	Rejected
SM6	7160.5	0.017	Hypothesis 6	Accepted
SM7	7974.5	0.323	Hypothesis 7	Rejected
SM8	7611.5	0.121	Hypothesis 8	Rejected
SM9	7771.0	0.175	Hypothesis 9	Rejected
SM10	6715.5	0.001	Hypothesis 10	Accepted
SM11	7203.0	0.015	Hypothesis 11	Accepted
SM12	7408.5	0.016	Hypothesis 12	Accepted
SM13	7223.5	0.002	Hypothesis 13	Accepted
SM14	6260.5	0.000	Hypothesis 14	Accepted
SM15	6796.5	0.002	Hypothesis 15	Accepted
SM16	6010.0	0.000	Hypothesis 16	Accepted
SM17	6684.5	0.001	Hypothesis 17	Accepted

Source: own elaboration.

## Data Availability

The data presented in this study are available on request from the corresponding author.

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
