# Peer review of "Lessons from the First Wave of COVID-19. What Security Measures Do Women and Men Require from the Hotel Industry to Protect against the Pandemic?"

_ijerph, 2021, doi:10.3390/ijerph18052232_

Round 1
Reviewer 1 Report
This paper is aimed at presenting the results of a study carried out in Spain to know the perceptions and opinions of travelers in touristic destinations which were impacted by this pandemic. Specifically, this research will address gender targeting, addressing the differences that may exist between women and men. This allows the tourism sector to obtain further information on women's opinions regarding the risks perceived during travel.
General opinion: The paper is overall well written, and it addresses an interesting and prominent topic.
However, some minor issues still require attention.
Particular remarks:
- A guided tour of the article is missing at the end of the Introduction.
- Page 7, row 320 – please correct the mistake in “who is employed or employed”.
- Why have not the authors considered the possibility of accounting for the different nationalities of costumers?
- The authors mention other studies. Is it possible to contrast the results obtained with other studies?
Author Response
Dear Reviewer, first of all, I wanted to thank you for your words and suggestions, without a doubt they make the manuscript have a higher quality.
In a specific way, I would like to tell you that the particular observations have been introduced in the manuscript. Especially the first, second and last of the observations. In relation to the latter we have included some studies that reinforce our research. As for nationalities, it is such a low value in which it contributes to the investigation that for that reason it has not been reflected. I hope this is not an inconvenience so that you can continue to consider the publishable manuscript.
Best regards.
Prof. Dr. Ramón Rueda López
Reviewer 2 Report
The issue of protective measures and protocols within a pandemic is certainly an important one. The promise of this research therefore is in the details that it can provide on some of the specific strategies that they can adopt to address the concerns for certain segments of the population. However, as presented/written, there are many areas of this manuscript that are in need of further review. These include:
Introduction-the introduction of the paper sets the stage for the nature of the pandemic and the distinctions with this health crises and those of others that preceded. However, there is a lack of attention to the specific gap that this research attempts to address and relatedly, a lack of critical engagement of the literature that speaks to this issue. While there is a hint of this in the title, this is not clearly presented within the introduction. This can be strengthened to set the tone and the potential relevance of this work.
Lack of contextualization-While the pandemic is a global concern, there is a lack of detail on the situation in Spain and the specific concerns that are being voiced at this time within this sector? Are there any media reports on this? Any other secondary data that underscores the concerns within this sector? Whether this is pre or post COVID, this will help set a context for this research and the interpretations of the findings as well.
Unclear Theoretical Framework-While there is a subsection on the theoretical framework, the section does not present this. In lieu of a discussion of a guiding theoretical perspective is a discussion on uncertainty and tourism. What is the framework here? Can you name this?
Review of the Literature and hypotheses: In the paper, the review of the literature cuts across areas of sustainability, gender differences and security measures in tourism. Immediately following these sections are a number of hypotheses that do not connect to the literature being presented here. This connection can serve as a justification for the relationships being tested and a way of situating the possible findings of this work. Relatedly, is it possible to create a scale for security measures and to test for differences between men and women being on a computed scale rather than individual questions within that scale? Also, based on the use of male/female divide, sex is being measured here not gender, which captures the social constructions around femininity and masculinity.
Methodology: Generally speaking, this section is well written. However, there are missing pieces of information around this, particularly the procedure for selecting participants and for sending out the online survey. It is important here to state the specific population or sample frame being targeted and the way in which persons were selected to participate in the study. In the paper, reference is made to probabilistic technical sampling with no specificity in the reporting.
Findings: Based on the findings, there is some reporting on the summative aspect of the data with analysis of differences in the mean scores across male and female participants. However, there is no analysis of the associations, if any, between sex (not gender) and perceptions of security measures implemented by the hoteliers. This is necessary to determine whether gender bears relevance to these perceptions, whether or not differences exist within these perceptions. This can also strengthen the work/paper.
Author Response
Dear Reviewer, first of all, I wanted to thank you for your words and suggestions, without a doubt they make the manuscript have a higher quality.
Specifically, I would like to tell you that the particular observations have been included in the manuscript:
1) Introduction-the introduction (...) This can be strengthened to set the tone and the potential relevance of this work. DONE
2) Lack of contextualization-While the pandemic (...) this will help set a context for this research and the interpretations of the findings as well. DONE
3) Unclear Theoretical Framework-While there is a (...) Can you name this?. DONE. I agree with you. For this reason we have eliminated the title "theoretical background" leaving only background information, we have also renamed the subsections. With this change I think the section makes better sense.
4) Review of the Literature and hypotheses: In the paper, (...) sex is being measured here not gender, which captures the social constructions around femininity and masculinity. DONE. I think that the global scale of measurement that has been proposed in the article called SRI, will clarify the aspect that you want to highlight, in any case, if not, please let me know. On the other hand, in the manuscript, we have rightly changed gender for sex.
5) Methodology: Generally speaking, (...) is made to probabilistic technical sampling with no specificity in the reporting. DONE
6) Findings: Based on the findings, there is some (...) This can also strengthen the work/paper. DONE. It is precisely this that he points out that we have highlighted in the limitations of this research. Specifically, one that has to do with having addressed only the part of the demand instead of having also studied the hotel offer.
I hope that the modifications introduced include everything that you wanted to convey to us with your observations and that, ultimately, they have improved the article in the terms you intended.
Best regards.
Prof. Dr. Ramón Rueda López
Reviewer 3 Report
The manuscript is interesting and of good quality, but can be improved in some points.
Authors should present the structure of the paper in the last paragraph of the introduction. In addition, authors should further emphasise why this work is relevant to its field of study.
The methodology must be developed, so that the work can be replicated in other future studies. What kind of analyses have been carried out and why these analyses and not others? They should better explain how to obtain the Safety Requirement Indicator (SRI). What is it and why is it important to obtain it? I think this needs to be looked at in more depth.
Table 1 should be more understandable at first sight.
The work lacks a discussion section and this is essential. "Authors should discuss the results and how they can be interpreted in perspective of previous studies and of the working hypotheses. The findings and their implications should be discussed in the broadest context possible".
Revise wording. There are words and expressions that are repeated too much, even in the same sentence.
Author Response
Dear Reviewer, first of all, I wanted to thank you for your words and suggestions, without a doubt they make the manuscript have a higher quality.
1) Authors should present (...) further emphasise why this work is relevant to its field of study. DONE
2) The methodology must be developed, (...) I think this needs to be looked at in more depth. DONE
3) Table 1 should be more understandable at first sight. DONE. Table 1 has been redesigned in a better way and some aspects that make it difficult to understand have been eliminated.
4) The work lacks a discussion section (...) be discussed in the broadest context possible". DONE. we have renamed section 4 and the discussion has been expanded taking as references the working hypotheses and previous studies.
5) Revise wording. DONE.
I hope that the modifications introduced include everything that you wanted to convey to us with your observations and that, ultimately, they have improved the article in the terms you intended.
Best regards.
Prof. Dr. Ramón Rueda López
Best regards.
Prof. Dr. Ramón Rueda López
Round 2
Reviewer 2 Report
This is a substantive improvement to the first version of this paper. I do agree with the removal of a theoretical section, and a replacement with a general background. There is more here as well on the process for collecting the data for this research. I note here that in sex was inserted throughout the paper in lieu of gender. While it is clear based on the instrument used for the research that the measure was for sex and not gender, gender can be retain in the literature, particularly where researchers would have applied more extensive and qualitative measures of these. Insert therefore where necessary in the literature. Otherwise, the author can insert a line, which indicates that while there is a holistic measure of the SMI, that individual hypotheses are presented for an item in the scale to test for the specific areas of differences based on sex. This will make clearer why the choice to include multiple hypotheses rather than one that speaks sex based differences based on the use a computed index score.
Author Response
Dear reviewer, thank you very much for your words.
Following your suggestion, we have incorporated the following paragraph in section 4.2.
"While the SRI allows, in a holistic way, to understand the level of demand for women and men in relation to the level of safety required against Covid-19, and whether there are differences between the sexes, at the same time propose a set of 17 individual hypothesis allows us to know in depth the safety measures in which women and men are most demanding and whether there are differences between the sexes".
I hope that this paragraph captures the meaning of your suggestion.
Sincerely
Dr. Ramon Rueda López